# Difference of Convex Functions Programming for Reinforcement Learning

**Bilal Piot**[1,2]**, Matthieu Geist**[1]**, Olivier Pietquin**[2,3]
[1]MaLIS research group (SUPELEC) - UMI 2958 (GeorgiaTech-CNRS), France
[2]LIFL (UMR 8022 CNRS/Lille 1) - SequeL team, Lille, France
[3] University Lille 1 - IUF (Institut Universitaire de France), France
bilal.piot@lifl.fr, matthieu.geist@supelec.fr, olivier.pietquin@univ-lille1.fr

### Abstract

Large Markov Decision Processes are usually solved using Approximate Dynamic Programming methods such as Approximate Value Iteration or Approximate Policy Iteration. The main contribution of this paper is to show that, alternatively, the optimal state-action value function can be estimated using Difference of Convex functions (DC) Programming. To do so, we study the minimization of a norm of the Optimal Bellman Residual (OBR) $T^*Q - Q$, where $T^*$ is the so-called optimal Bellman operator. Controlling this residual allows controlling the distance to the optimal action-value function, and we show that minimizing an empirical norm of the OBR is consistant in the Vapnik sense. Finally, we frame this optimization problem as a DC program. That allows envisioning using the large related literature on DC Programming to address the Reinforcement Leaning problem.

## 1   Introduction

This paper addresses the problem of solving large state-space Markov Decision Processes (MDPs)[16] in an infinite time horizon and discounted reward setting. The classical methods to tackle this problem, such as Approximate Value Iteration (AVI) or Approximate Policy Iteration (API) [6, 16][1], are derived from Dynamic Programming (DP). Here, we propose an alternative path. The idea is to search directly a function $Q$ for which $T^*Q \approx Q$, where $T^*$ is the optimal Bellman operator, by minimizing a norm of the Optimal Bellman Residual (OBR) $T^*Q - Q$. First, in Sec. 2.2, we show that the OBR Minimization (OBRM) is interesting, as it can serve as a proxy for the optimal action-value function estimation. Then, in Sec. 3, we prove that minimizing an empirical norm of the OBR is consistant in the Vapnick sense (this justifies working with sampled transitions). However, this empirical norm of the OBR is not convex. We hypothesize that this is why this approach is not studied in the literature (as far as we know), a notable exception being the work of Baird [5]. Therefore, our main contribution, presented in Sec. 4, is to show that this minimization can be framed as a minimization of a Difference of Convex functions (DC) [11]. Thus, a large literature on Difference of Convex functions Algorithms (DCA) [19, 20](a rather standard approach to non-convex programming) is available to solve our problem. Finally in Sec. 5, we conduct a generic experiment that compares a naive implementation of our approach to API and AVI methods, showing that it is competitive.

## 2 Background

### 2.1 MDP and ADP

Before describing the framework of MDPs in the infinite-time horizon and discounted reward setting, we give some general notations. Let $(\mathbb{R}, |.|)$ be the real space with its canonical norm and $X$ a finite set, $\mathbb{R}^X$ is the set of functions from $X$ to $\mathbb{R}$. The set of probability distributions over $X$ is noted $\Delta_X$. Let $Y$ be a finite set, $\Delta_X^Y$ is the set of functions from $Y$ to $\Delta_X$. Let $\alpha \in \mathbb{R}^X$, $p \geq 1$ and $\nu \in \Delta_X$, we define the $\mathbf{L}_{p,\nu}$-semi-norm of $\alpha$, noted $\|\alpha\|_{p,\nu}$, by: $\|\alpha\|_{p,\nu} = (\sum_{x \in X} \nu(x)|\alpha(x)|^p)^{\frac{1}{p}}$. In addition, the infinite norm is noted $\|\alpha\|_\infty$ and defined as $\|\alpha\|_\infty = \max_{x \in X} |\alpha(x)|$. Let $v$ be a random variable which takes its values in $X$, $v \sim \nu$ means that the probability that $v = x$ is $\nu(x)$.

Now, we provide a brief summary of some of the concepts from the theory of MDP and ADP [16]. Here, the agent is supposed to act in a finite MDP [2] represented by a tuple $M = \{S, A, R, P, \gamma\}$ where $S = \{s_i\}_{1 \leq i \leq N_S}$ is the state space, $A = \{a_i\}_{1 \leq i \leq N_A}$ is the action space, $R \in \mathbb{R}^{S \times A}$ is the reward function, $\gamma \in ]0,1[$ is a discount factor and $P \in \Delta_S^{S \times A}$ is the Markovian dynamics which gives the probability, $P(s'|s,a)$, to reach $s'$ by choosing action $a$ in state $s$. A policy $\pi$ is an element of $A^S$ and defines the behavior of an agent. The quality of a policy $\pi$ is defined by the action-value function. For a given policy $\pi$, the action-value function $Q^\pi \in \mathbb{R}^{S \times A}$ is defined as $Q^\pi(s,a) = \mathbb{E}^\pi[\sum_{t=0}^{+\infty} \gamma^t R(s_t, a_t)]$, where $\mathbb{E}^\pi$ is the expectation over the distribution of the admissible trajectories $(s_0, a_0, s_1, \pi(s_1), \dots)$ obtained by executing the policy $\pi$ starting from $s_0 = s$ and $a_0 = a$. Moreover, the function $Q^* \in \mathbb{R}^{S \times A}$ defined as $Q^* = \max_{\pi \in A^S} Q^\pi$ is called the optimal action-value function. A policy $\pi$ is optimal if $\forall s \in S, Q^\pi(s, \pi(s)) = Q^*(s, \pi(s))$. A policy $\pi$ is said greedy with respect to a function $Q$ if $\forall s \in S, \pi(s) \in \operatorname{argmax}_{a \in A} Q(s, a)$. Greedy policies are important because a policy $\pi$ greedy with respect to $Q^*$ is optimal. In addition, as we work in the finite MDP setting, we define, for each policy $\pi$, the matrix $P_\pi$ of size $N_S N_A \times N_S N_A$ with elements $P_\pi((s,a),(s',a')) = P(s'|s,a)\mathbf{1}_{\{\pi(s')=a'\}}$. Let $\nu \in \Delta_{S \times A}$, we note $\nu P_\pi \in \Delta_{S \times A}$ the distribution such that $(\nu P_\pi)(s,a) = \sum_{(s',a') \in S \times A} \nu(s',a') P_\pi((s',a'),(s,a))$. Finally, $Q^\pi$ and $Q^*$ are known to be fixed points of the contracting operators $T^\pi$ and $T^*$ respectively:

$$\forall Q \in \mathbb{R}^{S \times A}, \forall (s,a) \in S \times A, \quad T^\pi Q(s,a) = R(s,a) + \gamma \sum_{s' \in S} P(s'|s,a) Q(s, \pi(s')),$$

$$\forall Q \in \mathbb{R}^{S \times A}, \forall (s,a) \in S \times A, \quad T^* Q(s,a) = R(s,a) + \gamma \sum_{s' \in S} P(s'|s,a) \max_{b \in A} Q(s,b).$$

When the state space becomes large, two important problems arise to solve large MDPs. The first one, called the *representation* problem, is that an exact representation of the values of the action-value functions is impossible, so these functions need to be represented with a moderate number of coefficients. The second problem, called the *sample* problem, is that there is no direct access to the Bellman operators but only samples from them. One solution for the representation problem is to linearly parameterize the action-value functions thanks to a basis of $d \in \mathbb{N}^*$ functions $\phi = (\phi_i)_{i=1}^d$ where $\phi_i \in \mathbb{R}^{S \times A}$. In addition, we define for each state-action couple $(s,a)$ the vector $\phi(s,a) \in \mathbb{R}^d$ such that $\phi(s,a) = (\phi_i(s,a))_{i=1}^d$. Thus, the action-value functions are characterized by a vector $\theta \in \mathbb{R}^d$ and noted $Q_\theta$ :

$$\forall \theta \in \mathbb{R}^d, \forall (s,a) \in S \times A, Q_\theta(s,a) = \sum_{i=1}^d \theta_i \phi_i(s,a) = \langle \theta, \phi(s,a) \rangle,$$

where $\langle .,. \rangle$ is the canonical dot product of $\mathbb{R}^d$.

The usual framework to solve large MDPs are for instance AVI and API. AVI consists in processing a sequence $(Q_{\theta_n}^{\text{AVI}})_{n \in \mathbb{N}}$ where $\theta_0 \in \mathbb{R}^d$ and $\forall n \in \mathbb{N}, Q_{\theta_{n+1}}^{\text{AVI}} \approx T^* Q_{\theta_n}^{\text{AVI}}$. API consists in processing two sequences $(Q_{\theta_n}^{\text{API}})_{n \in \mathbb{N}}$ and $(\pi_n^{\text{API}})_{n \in \mathbb{N}}$ where $\pi_0^{\text{API}} \in A^S$, $\forall n \in \mathbb{N}, Q_{\theta_n}^{\text{API}} \approx$

$T^{\pi_n} Q^{\text{API}}_{\theta_n}$ and $\pi^{\text{API}}_{n+1}$ is greedy with respect to $Q^{\text{API}}_{\theta_n}$. The approximation steps in AVI and API generate the sequences of errors $(\epsilon^{\text{AVI}}_n = T^* Q^{\text{AVI}}_{\theta_n} - Q^{\text{AVI}}_{\theta_{n+1}})_{n \in \mathbb{N}}$ and $(\epsilon^{\text{API}}_n = T^{\pi_n} Q^{\text{API}}_{\theta_n} - Q^{\text{API}}_{\theta_n})_{n \in \mathbb{N}}$ respectively. Those approximation errors are due to both the representation and the sample problems and can be made explicit for specific implementations of those methods [14, 1]. These ALP methods are legitimated by the following bound [15, 9]:

$$\limsup_{n \to \infty} \|Q^* - Q^{\pi^{\text{API}\backslash\text{AVI}}_n}\|_{p,\nu} \leq \frac{2\gamma}{(1-\gamma)^2} C_2(\nu,\mu)^{\frac{1}{p}} \epsilon^{\text{API}\backslash\text{AVI}}, \tag{1}$$

where $\pi^{\text{API}\backslash\text{AVI}}_n$ is greedy with respect to $Q^{\text{API}\backslash\text{AVI}}_{\theta_n}$, $\epsilon^{\text{API}\backslash\text{AVI}} = \sup_{n \in \mathbb{N}} \|\epsilon^{\text{API}\backslash\text{AVI}}_n\|_{p,\mu}$ and $C_2(\nu,\mu)$ is a second order concentrability coefficient, $C_2(\nu,\mu) = (1-\gamma) \sum_{m \geq 1} m \gamma^{m-1} c(m)$, where $c(m) = \max_{\pi_1,\ldots,\pi_m,(s,a) \in S \times A} \frac{(\nu P_{\pi_1} P_{\pi_2} \ldots P_{\pi_m})(s,a)}{\mu(s,a)}$. In the next section, we compare the bound Eq. (1) with a similar bound derived from the OBR minimization approach in order to justify it.

## 2.2 Why minimizing the OBR?

The aim of Dynamic Programming (DP) is, given an MDP $M$, to find $Q^*$ which is equivalent to minimizing a certain norm of the OBR $J_{p,\mu}(Q) = \|T^*Q - Q\|_{p,\mu}$ where $\mu \in \Delta_{S \times A}$ is such that $\forall (s,a) \in S \times A, \mu(s,a) > 0$ and $p \geq 1$. Indeed, it is trivial to verify that the only minimizer of $J_{p,\mu}$ is $Q^*$. Moreover, we have the following bound given by Th. 1.

**Theorem 1.** *Let $\nu \in \Delta_{S \times A}$, $\mu \in \Delta_{S \times A}$, $\hat{\pi} \in A^S$ and $C_1(\nu,\mu,\hat{\pi}) \in [1,+\infty[\cup\{+\infty\}$ the smallest constant verifying $(1-\gamma)\nu \sum_{t \geq 0} \gamma^t P^t_{\hat{\pi}} \leq C_1(\nu,\mu,\hat{\pi})\mu$, then:*

$$\forall Q \in \mathbb{R}^{S \times A}, \|Q^* - Q^\pi\|_{p,\nu} \leq \frac{2}{1-\gamma} \left( \frac{C_1(\nu,\mu,\pi) + C_1(\nu,\mu,\pi^*)}{2} \right)^{\frac{1}{p}} \|T^*Q - Q\|_{p,\mu}, \tag{2}$$

*where $\pi$ is greedy with respect to $Q$ and $\pi^*$ is any optimal policy.*

*Proof.* A proof is given in the supplementary file. Similar results exist [15]. $\square$

In Reinforcement Leaning (RL), because of the representation and the sample problems, minimizing $\|T^*Q - Q\|_{p,\mu}$ over $\mathbb{R}^{S \times A}$ is not possible (see Sec. 3 for details), but we can consider that our approach provides us a function $Q$ such that $T^*Q \approx Q$ and define the error $\epsilon^{\text{OBRM}} = \|T^*Q - Q\|_{p,\mu}$. Thus, via Eq. (2), we have:

$$\|Q^* - Q^\pi\|_{p,\nu} \leq \frac{2}{1-\gamma} \left( \frac{C_1(\nu,\mu,\pi) + C_1(\nu,\mu,\pi^*)}{2} \right)^{\frac{1}{p}} \epsilon^{\text{OBRM}}, \tag{3}$$

where $\pi$ is greedy with respect to $Q$. This bound has the same form as the one of API and AVI described in Eq. (1) and the Tab. 1 allows comparing them. This bound has two

| Algorithms | Horizon term | Concentrability term | Error term |
|---|---|---|---|
| API\AVI | $\frac{2\gamma}{(1-\gamma)^2}$ | $C_2(\nu,\mu)$ | $\epsilon^{\text{API}\backslash\text{AVI}}$ |
| OBRM | $\frac{2}{1-\gamma}$ | $\frac{C_1(\nu,\mu,\pi)+C_1(\nu,\mu,\pi^*)}{2}$ | $\epsilon^{\text{OBRM}}$ |

Table 1: Bounds comparison.

advantages over API\AVI. First, the horizon term $\frac{2}{1-\gamma}$ is better than the horizon term $\frac{2\gamma}{(1-\gamma)^2}$ as long as $\gamma > 0.5$, which is the usual case. Second, the concentrability term $\frac{C_1(\nu,\mu,\pi)+C_1(\nu,\mu,\pi^*)}{2}$ is considered better that $C_2(\nu,\mu)$, mainly because if $C_2(\nu,\mu) < +\infty$ then $\frac{C_1(\nu,\mu,\pi)+C_1(\nu,\mu,\pi^*)}{2} < +\infty$, the contrary being not true (see [17] for a discussion about the comparison of these concentrability coefficients). Thus, the bound Eq. (3) justifies the minimization of a norm of the OBR, as long as we are able to control the error term $\epsilon^{\text{OBRM}}$.

## 3 Vapnik-Consistency of the empirical norm of the OBR

When the state space is too large, it is not possible to minimize directly $\|T^*Q - Q\|_{p,\mu}$, as we need to compute $T^*Q(s,a)$ for each couple $(s,a)$ (sample problem). However, we can consider the case where we choose $N$ samples represented by $N$ independent and identically distributed random variables $(S_i, A_i)_{1 \leq i \leq N}$ such that $(S_i, A_i) \sim \mu$ and minimize $\|T^*Q - Q\|_{p,\mu_N}$ where $\mu_N$ is the empirical distribution $\mu_N(s,a) = \frac{1}{N} \sum_{i=1}^{N} \mathbf{1}_{\{(S_i, A_i) = (s,a)\}}$. An important question (answered below) is to know if controlling the empirical norm allows controlling the true norm of interest (consistency in the Vapnik sense [22]), and at what rate convergence occurs.

Computing $\|T^*Q - Q\|_{p,\mu_N} = (\frac{1}{N} \sum_{i=1}^{N} |T^*Q(S_i, A_i) - Q(S_i, A_i)|^p)^{\frac{1}{p}}$ is tractable if we consider that we can compute $T^*Q(S_i, A_i)$ which means that we have a perfect knowledge of the dynamics $P$ and that the number of next states for the state-action couple $(S_i, A_i)$ is not too large. In Sec. 4.3, we propose different solutions to evaluate $T^*Q(S_i, A_i)$ when the number of next states is too large or when the dynamics is not provided. Now, the natural question is to what extent minimizing $\|T^*Q - Q\|_{p,\mu_N}$ corresponds to minimizing $\|T^*Q - Q\|_{p,\mu}$. In addition, we cannot minimize $\|T^*Q - Q\|_{p,\mu_N}$ over $\mathbb{R}^{S \times A}$ as this space is too large (representation problem) but over the space $\{Q_\theta \in \mathbb{R}^{S \times A}, \theta \in \mathbb{R}^d\}$. Moreover, as we are looking for a function such that $Q_\theta = Q^*$, we can limit our search to the functions satisfying $\|Q_\theta\|_\infty \leq \frac{\|R\|_\infty}{1-\gamma}$. Thus, we search for a function $Q$ in the hypothesis space $\mathfrak{Q} = \{Q_\theta \in \mathbb{R}^{S \times A}, \theta \in \mathbb{R}^d, \|Q_\theta\|_\infty \leq \frac{\|R\|_\infty}{1-\gamma}\}$, in order to minimize $\|T^*Q - Q\|_{p,\mu_N}$. Let $Q_N \in \text{argmin}_{Q \in \mathfrak{Q}} \|T^*Q - Q\|_{p,\mu_N}$ be a minimizer of the empirical norm of the OBR, we want to know to what extent the empirical error $\|T^*Q_N - Q_N\|_{p,\mu_N}$ is related to the real error $\epsilon^{\text{OBRM}} = \|T^*Q_N - Q_N\|_{p,\mu}$. The answer for deterministic-finite MPDs relies in Th. 2 (the continuous-stochastic MDP case being discussed shortly after).

**Theorem 2.** *Let* $\eta \in ]0,1[$ *and* $M$ *be a finite deterministic MDP, with probability at least* $1 - \eta$*, we have:*

$$\forall Q \in \mathfrak{Q}, \|T^*Q - Q\|_{p,\mu}^p \leq \|T^*Q - Q\|_{p,\mu_N}^p + \frac{2\|R\|_\infty}{1-\gamma} \sqrt{\varepsilon(N)},$$

*where* $\varepsilon(N) = \frac{h(ln(\frac{2N}{h})+1)+ln(\frac{4}{\eta})}{N}$ *and* $h = 2N_A(d+1)$*. With probability at least* $1 - 2\eta$*:*

$$\epsilon^{OBRM} = \|T^*Q_N - Q_N\|_{p,\mu}^p \leq \epsilon^B + \frac{2\|R\|_\infty}{1-\gamma} \left( \sqrt{\varepsilon(N)} + \sqrt{\frac{ln(1/\eta)}{2N}} \right),$$

*where* $\epsilon^B = \min_{Q \in \mathfrak{Q}} \|T^*Q - Q\|_{p,\mu}^p$ *is the error due to the choice of features.*

*Proof.* The complete proof is provided in the supplementary file. It mainly consists in computing the Vapnik-Chervonenkis dimension of the residual. □

Thus, if we were able to compute a function such as $Q_N$, we would have, thanks to Eq .(2) and Th. 2:

$$\|Q^* - Q^{\pi_N}\|_{p,\nu} \leq \left( \frac{C_1(\nu,\mu,\pi_N) + C_1(\nu,\mu,\pi^*)}{1-\gamma} \right)^{\frac{1}{p}} \left( \epsilon^B + \frac{2\|R\|_\infty}{1-\gamma} \left( \sqrt{\varepsilon(N)} + \sqrt{\frac{ln(1/\eta)}{2N}} \right) \right)^{\frac{1}{p}}.$$

where $\pi_N$ is greedy with respect to $Q_N$. The error term $\epsilon^{\text{OBRM}}$ is explicitly controlled by two terms $\epsilon^B$, a term of bias, and $\frac{2\|R\|_\infty}{1-\gamma} \left( \sqrt{\varepsilon(N)} + \sqrt{\frac{ln(1/\eta)}{2N}} \right)$ a term of variance. The term $\epsilon^B = \min_{Q \in \mathfrak{Q}} \|T^*Q - Q\|_{p,\mu}^p$ is relative to the representation problem and is fixed by the choice of features. The term of variance is decreasing at the speed $\sqrt{\frac{1}{N}}$.

A similar bound can be obtained for non-deterministic continuous-state MDPs with finite number of actions where the state space is a compact set in a metric space, the features

$(\phi_i)_{i=1}^d$ are Lipschitz and for each state-action couple the next states belongs to a ball of fixed radius. The proof is a simple extension of the one given in the supplementary material. Those continuous MDPs are representative of real dynamical systems. Now that we know that minimizing $\|T^*Q - Q\|_{p,\mu_N}^p$ allows controlling $\|Q^* - Q^{\pi_N}\|_{p,\nu}$, the question is how do we frame this optimization problem. Indeed $\|T^*Q - Q\|_{p,\mu_N}^p$ is a non-convex and a non-differentiable function with respect to $Q$, thus a direct minimization could lead us to bad solutions. In the next section, we propose a method to alleviate those difficulties.

## 4 Reduction to a DC problem

Here, we frame the minimization of the empirical norm of the OBR as a DC problem and instantiate a general algorithm, DCA [20], that tries to solve it. First, we provide a short introduction to difference of convex functions.

### 4.1 DC background

Let $E$ be a finite dimensional Hilbert space and $\langle .,. \rangle_E$, $\|.\|_E$ its dot product and norm respectively. We say that a function $f \in \mathbb{R}^E$ is DC if there exists $g, h \in \mathbb{R}^E$ which are convex and lower semi-continuous such that $f = g - h$. The set of DC functions is noted $DC(E)$ and is stable to most of the operations that can be encountered in optimization, contrary to the set of convex functions. Indeed, let $(f_i)_{i=1}^K$ be a sequence of $K \in \mathbb{N}^*$ DC functions and $(\alpha_i)_{i=1}^K \in \mathbb{R}^K$ then $\sum_{i=1}^K \alpha_i f_i$, $\prod_{i=1}^K f_i$, $\min_{1 \leq i \leq K} f_i$, $\max_{1 \leq i \leq K} f_i$ and $|f_i|$ are DC functions [11]. In order to minimize a DC function $f = g - h$, we need to define a notion of differentiability for convex and lower semi-continuous functions. Let $g$ be such a function and $e \in E$, we define the sub-gradient $\partial_e g$ of $g$ in $e$ as:

$$\partial_e g = \{\delta \in E, \forall e' \in E, g(e') \geq g(e) + \langle e' - e, \delta \rangle_E\}.$$

For a convex and lower semi-continuous $g \in \mathbb{R}^E$, the sub-gradient $\partial_e g$ is non empty for all $e \in E$ [11]. This observation leads to a minimization method of a function $f \in DC(E)$ called Difference of Convex functions Algorithm (DCA). Indeed, as $f$ is DC, we have:

$$\forall (e, e') \in E^2, f(e') = g(e') - h(e') \underset{(a)}{\leq} g(e') - h(e) - \langle e' - e, \delta \rangle_E,$$

where $\delta \in \partial_e h$ and inequality $(a)$ is true by definition of the sub-gradient. Thus, for all $e \in E$, the function $f$ is upper bounded by a function $f_e \in \mathbb{R}^E$ defined for all $e' \in E$ by $f_e(e') = g(e') - h(e) - \langle e' - e, \delta \rangle_E$. The function $f_e$ is a convex and lower semi-continuous function (as it is the sum of two convex and lower semi-continuous functions which are $g$ and the linear function $\forall e' \in E, \langle e - e', \delta \rangle_E - h(e)$). In addition, those functions have the particular property that $\forall e \in E, f(e) = f_e(e)$. The set of convex functions $(f_e)_{e \in E}$ that upper-bound the function $f$ plays a key role in DCA.

The algorithm DCA [20] consists in constructing a sequence $(e_n)_{n \in \mathbb{N}}$ such that the sequence $(f(e_n))_{n \in \mathbb{N}}$ decreases. The first step is to choose a starting point $e_0 \in E$, then we minimize the convex function $f_{e_0}$ that upper-bounds the function $f$. We note $e_1$ a minimizer of $f_{e_0}$, $e_1 \in \text{argmin}_{e \in E} f_{e_0}$. This minimization can be realized by any convex optimization solver. As $f(e_0) = f_{e_0}(e_0) \geq f_{e_0}(e_1)$ and $f_{e_0}(e_1) \geq f(e_1)$, then $f(e_0) \geq f(e_1)$. Thus, if we construct the sequence $(e_n)_{n \in \mathbb{N}}$ such that $\forall n \in \mathbb{N}, e_{n+1} \in \text{argmin}_{e \in E} f_{e_n}$ and $e_0 \in E$, then we obtain a decreasing sequence $(f(e_n))_{n \in \mathbb{N}}$. Therefore, the algorithm DCA solves a sequence of convex optimization problems in order to solve a DC optimization problem. Three important choices can radically change the DCA performance: the first one is the explicit choice of the decomposition of $f$, the second one is the choice of the starting point $e_0$ and finally the choice of the intermediate convex solver. The DCA algorithm hardly guarantee convergence to the global optima, but it usually provides good solutions. Moreover, it has some nice properties when one of the functions $g$ or $h$ is polyhedral. A function $g$ is said polyhedral when $\forall e \in E, g(e) = \max_{1 \leq i \leq K}[\langle \alpha_i, e \rangle_H + \beta_i]$, where $(\alpha_i)_{i=1}^K \in E^K$ and $(\beta_i)_{i=1}^K \in \mathbb{R}^K$. If one of the function $g, h$ is polyhedral, $f$ is under bounded and the DCA sequence $(e_n)_{n \in \mathbb{N}}$ is bounded, the DCA algorithm converges in finite time to a local minima. The finite time aspect is quite interesting in term of implementation. More details about DC programming and DCA are given in [20] and even conditions for convergence to the global optima.

## 4.2 The OBR minimization framed as a DC problem

A first important result is that for any choice of $p \geq 1$, the OBRM is actually a DC problem.

**Theorem 3.** *Let $J_{p,\mu_N}^p(\theta) = \|T^*Q_\theta - Q_\theta\|_{p,\mu_N}$ be a function from $\mathbb{R}^d$ to reals, $J_{p,\mu_N}^p(\theta)$ is a DC functions when $p \in \mathbb{N}^*$.*

*Proof.* Let us write $J_{p,\mu_N}^p$ as:

$$J_{p,\mu_N}^p(\theta) = \frac{1}{N}\sum_{i=1}^{N}|\langle\phi(S_i,A_i),\theta\rangle - R(S_i,A_i) - \gamma\sum_{s'\in S}P(s'|S_i,A_i)\max_{a\in A}\langle\phi(s',a),\theta\rangle|^p.$$

First, as for each $(S_i, A_i)$ the linear function $\langle\phi(S_i,A_i),.\rangle$ is convex and continuous, the affine function $g_i = \langle\phi(S_i,A_i),.\rangle + R(S_i,A_i)$ is convex and continuous. Therefore, the function $\max_{a\in A}\langle\phi(s',a),.\rangle$ is also convex and continuous as a finite maximum of convex and continuous functions. In addition, the function $h_i = \gamma\sum_{s'\in S}P(s'|S_i,A_i)\max_{a\in A}\langle\phi(s',a),.\rangle|$ is convex and continuous as a positively weighted finite sum of convex and continuous functions. Thus, the function $f_i = g_i - h_i$ is a DC function. As an absolute value of a DC function is DC, a finite product of DC functions is DC and a weighted sum of DC functions is DC, then $J_{p,\mu_N}^p = \frac{1}{N}\sum_{i=1}^{N}|f_i|^p$ is a DC function. $\square$

However, knowing that $J_{p,\mu_N}^p$ is DC is not sufficient in order to use the DCA algorithm. Indeed, we need an explicit decomposition of $J_{p,\mu_N}^p$ as a difference of two convex functions. We present two polyhedral explicit decompositions of $J_{p,\mu_N}^p$ when $p = 1$ and when $p = 2$.

**Theorem 4.** *There exists explicit polyhedral decompositions of $J_{p,\mu_N}^p$ when $p = 1$ and $p = 2$.*

*For $p = 1$: $J_{1,\mu_N} = G_{1,\mu_N} - H_{1,\mu_N}$, where $G_{1,\mu_N} = \frac{1}{N}\sum_{i=1}^{N}2\max(g_i,h_i)$ and $H_{1,\mu_N} = \frac{1}{N}\sum_{i=1}^{N}(g_i + h_i)$, with $g_i = \langle\phi(S_i,A_i),.\rangle + R(S_i,A_i)$ and $h_i = \gamma\sum_{s'\in S}P(s'|S_i,A_i)\max_{a\in A}\langle\phi(s',a),.\rangle$.*

*For $p = 2$: $J_{2,\mu_N}^2 = G_{2,\mu_N} - H_{2,\mu_N}$, where $G_{2,\mu_N} = \frac{1}{N}\sum_{i=1}^{N}[\bar{g}_i^2 + \bar{h}_i^2]$ and $H_{2,\mu_N} = \frac{1}{N}\sum_{i=1}^{N}[\bar{g}_i + \bar{h}_i]^2$ with:*

$$\bar{g}_i = \max(g_i,h_i) + g_i - \left(\langle\phi(S_i,A_i) + \gamma\sum_{s'\in S}P(s'|S_i,A_i)\phi(s',a_1),.\rangle - R(S_i,A_i)\right),$$

$$\bar{h}_i = \max(g_i,h_i) + h_i - \left(\langle\phi(S_i,A_i) + \gamma\sum_{s'\in S}P(s'|S_i,A_i)\phi(s',a_1),.\rangle - R(S_i,A_i)\right).$$

*Proof.* The proof is provided in the supplementary material. $\square$

Unfortunately, there is currently no guarantee that DCA applied to $J_{p,\mu_N}^p = G_{p,\mu_N} - H_{p,\mu_N}$ outputs $Q_N \in \arg\min_{Q\in\mathfrak{Q}}\|T^*Q - Q\|_{p,\mu_N}$. The error between the output $\hat{Q}_N$ of DCA and $Q_N$ is not studied here but it is a nice theoretical perspective for future works.

## 4.3 The batch scenario

Previously, we admit that it was possible to calculate $T^*Q(s,a) = R(s,a) + \gamma\sum_{s'\in S}P(s'|s,a)\max_{b\in A}Q(s',b)$. However, if the number of next states $s'$ for a given couple $(s,a)$ is too large or if $T^*$ is unknown, this can be intractable. A solution, when we have a simulator, is to generate for each couple $(S_i, A_i)$ a set of $N'$ samples $(S'_{i,j})_{j=1}^{N'}$ and provide a non-biased estimation of $T^*Q(S_i,A_i)$: $\hat{T}^*Q(S_i,A_i) = R(S_i,A_i) + \gamma\frac{1}{N'}\sum_{j=1}^{N'}\max_{a\in A}Q(S'_{i,j},a)$. Even if $|\hat{T}^*Q(S_i,a_i) - Q(S_i,A_i)|^p$ is a biased estimator of $|T^*Q(S_i,A_i) - Q(S_i,A_i)|^p$, this biais can be controlled by the number of samples $N'$.

In the case where we do not have such a simulator, but only sampled transitions $(S_i, A_i, S'_i)_{i=1}^{N}$ (the batch scenario), it is possible to provide a non-biased estimation of

$T^*Q(S_i, A_i)$ via: $\hat{T}^*Q(S_i, A_i) = R(S_i, A_i) + \gamma \max_{b \in A} Q(S'_i, b)$. However in that case, $|\hat{T}^*Q(S_i, A_i) - Q(S_i, A_i)|^p$ is a biased estimator of $|T^*Q(S_i, A_i) - Q(S_i, A_i)|^p$ and the biais is uncontrolled [2]. In order to alleviate this typical problem from the batch scenario, several techniques have been proposed in the literature to provide a better estimator $|\hat{T}^*Q(S_i, A_i) - Q(S_i, A_i)|^p$, such as embeddings in Reproducing Kernel Hilbert Spaces (RKHS)[13] or locally weighted averager such as Nadaraya-Watson estimators[21]. In both cases, the non-biased estimation of $T^*Q(S_i, A_i)$ takes the form $\hat{T}^*Q(S_i, A_i) = R(S_i, A_i) + \gamma \frac{1}{N} \sum_{j=1}^{N} \beta_i(S'_j) \max_{a \in A} Q(S'_j, a)$, where $\beta_i(S'_j)$ represents the weight of the samples $S'_j$ in the estimation of $T^*Q(S_i, A_i)$. To obtain an explicit DC decomposition, when $p = 1$ or $p = 2$, of $\hat{J}^p_{p,\mu_N}(\theta) = \frac{1}{N} \sum_{i=1}^{N} |\hat{T}^*Q_\theta(S_i, A_i) - Q_\theta(S_i, A_i)|^p$ it is sufficient to replace $\sum_{s' \in S} P(s'|S_i, A_i) \max_{a \in A} \langle \phi(s', a), \theta \rangle$ by $\frac{1}{N} \sum_{j=1}^{N} \beta_i(S'_j) \max_{a \in A} Q(S'_j, a)$ (or $\frac{1}{N'} \sum_{j=1}^{N'} \max_{a \in A} Q(S'_{i,j}, a)$ if we have a simulator) in the DC decomposition of $J^p_{p,\mu_N}$.

## 5 Illustration

This experiment focuses on stationary Garnet problems, which are a class of randomly constructed finite MDPs representative of the kind of finite MDPs that might be encountered in practice [3]. A stationary Garnet problem is characterized by 3 parameters: $Garnet(N_S, N_A, N_B)$. The parameters $N_S$ and $N_A$ are the number of states and actions respectively, and $N_B$ is a branching factor specifying the number of next states for each state-action pair. Here, we choose a particular type of Garnets which presents a topological structure relative to real dynamical systems and aims at simulating the behavior of a smooth continuous-state MDPs (as described in Sec. 3). Those systems are generally multi-dimensional state spaces MDPs where an action leads to different next states close to each other. The fact that an action leads to close next states can model the noise in a real system for instance. Thus, problems such as the highway simulator [12], the mountain car or the inverted pendulum (possibly discretized) are particular cases of this type of Garnets. For those particular Garnets, the state space is composed of $d$ dimensions ($d = 2$ in this particular experiment) and each dimension $i$ has a finite number of elements $x_i$ ($x_i = 10$). So, a state $s = [s^1, s^2, .., s^i, .., s^d]$ is a $d$-uple where each composent $s^i$ can take a finite value between 1 and $x_i$. In addition, the distance between two states $s, s'$ is $\|s - s'\|^2 = \sum_{i=1}^{i=d}(s^i - s'^i)^2$. Thus, we obtain MDPs with a state space size of $\prod_{i=1}^{d} x_i$. The number of actions is $N_A = 5$. For each state action couple $(s, a)$, we choose randomly $N_B$ next states ($N_B = 5$) via a Gaussian distribution of $d$ dimensions centered in $s$ where the covariance matrix is the identity matrix of size $d$, $I_d$, multiply by a term $\sigma$ (here $\sigma = 1$). This allows handling the smoothness of the MDP: if $\sigma$ is small the next states $s'$ are close to $s$ and if $\sigma$ is large, the next states $s'$ can be very far form each other and also from $s$. The probability of going to each next state $s'$ is generated by partitioning the unit interval at $N_B - 1$ cut points selected randomly. For each couple $(s, a)$, the reward $R(s, a)$ is drawn uniformly between $-1$ and $1$. For each Garnet problem, it is possible to compute an optimal policy $\pi^*$ thanks to the policy iteration algorithm.

In this experiment, we construct 50 Garnets $\{G_p\}_{1 \le p \le 50}$ as explained before. For each Garnet $G_p$, we build 10 data sets $\{D^{p,q}\}_{1 \le q \le 10}$ composed of $N$ sampled transitions $(s_i, a_i, s'_i)_{i=1}^{N}$ drawn uniformly and independently. Thus, we are in the batch scenario. The minimization of $J_{1,N}$ and $J_{2,N}$ via the DCA algorithms, where the estimation of $T^*Q(s_i, a_i)$ is done via $R(s_i, a_i) + \gamma \max_{b \in A} Q(s'_i, b)$ (so uncontrolled biais), are called DCA$_1$ and DCA$_2$ respectively. The initialisation of DCA is $\theta_0 = 0$ and the intermediary optimization convex problems are solved by a sub-gradient descent [18]. Those two algorithms are compared with state-of the art Reinforcement Learning algorithms which are LSPI (API implementation) and Fitted-Q (AVI implementation). The four algorithms uses the tabular basis. Each algorithm outputs a function $Q_A^{p,q} \in \mathbb{R}^{S \times A}$ and the policy associated to $Q_A^{p,q}$ is $\pi_A^{p,q}(s) = \operatorname{argmax}_{a \in A} Q_A^{p,q}(s, a)$. In order to quantify the performance of a given algorithm, we calculate the criterion $T_A^{p,q} = \frac{\mathbb{E}_\rho[V^{\pi^*} - V^{\pi_A^{p,q}}]}{\mathbb{E}_\rho[|V^{\pi^*}|]}$, where $V^{\pi_A^{p,q}}$ is computed via the policy evaluation algorithm. The mean performance criterion $T_A$ is $\frac{1}{500} \sum_{p=1}^{50} \sum_{q=1}^{10} T_A^{p,q}$. We also

calculate, for each algorithm, the variance criterion $\mathrm{std}_A^p = \frac{1}{10}\sum_{q=1}^{10}(T_A^{p,q} - \frac{1}{10}\sum_{q=1}^{10} T_A^{p,q})^2$ and the resulting mean variance criterion is $\mathrm{std}_A = \frac{1}{50}\sum_{p=1}^{50}\mathrm{std}_A^p$. In Fig. 1(a), we plot the performance versus the number of samples. We observe that the 4 algorithms have similar performances, which shows that our alternative approach is competitive. In Fig. 1(b), we

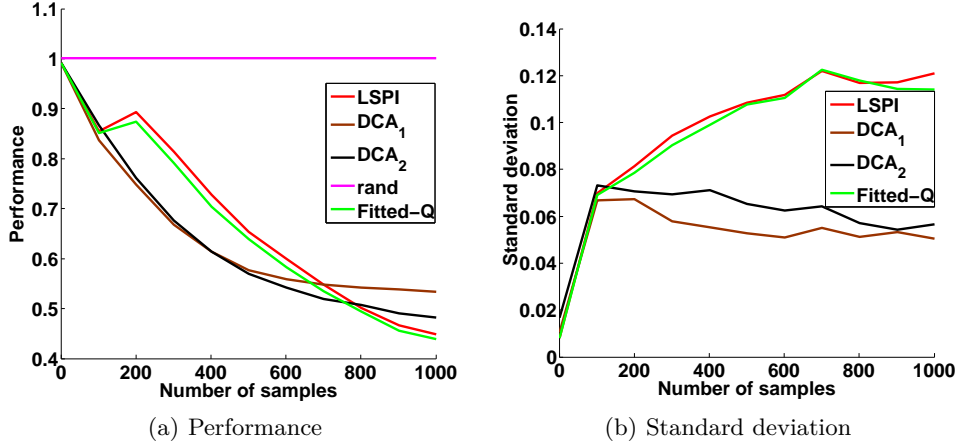

(a) Performance            (b) Standard deviation

Figure 1: Garnet Experiment

plot the standard deviation versus the number of samples. Here, we observe that DCA algorithms have less variance which is an advantage. This experiment shows us that DC programming is relevant for RL but still has to prove its efficiency on real problems.

## 6 Conclusion and Perspectives

In this paper, we presented an alternative approach to tackle the problem of solving large MDPs by estimating the optimal action-value function via DC Programming. To do so, we first showed that minimizing a norm of the OBR is interesting. Then, we proved that the empirical norm of the OBR is consistant in the Vapnick sense (strict consistency). Finally, we framed the minimization of the empirical norm as DC minimization which allows us to rely on the literature on DCA. We conduct a generic experiment with a basic setting for DCA as we choose a canonical explicit decomposition of our DC functions criterion and a sub-gradient descent in order to minimize the intermediary convex minimization problems. We obtain similar results to AVI and API. Thus, an interesting perspective would be to have a less naive setting for DCA by choosing different explicit decompositions and find a better convex solver for the intermediary convex minimizations problems. Another interesting perspective is that our approach can be non-parametric. Indeed, as pointed in [10] a convex minimization problem can be solved via boosting techniques which avoids the choice of features. Therefore, each intermediary convex problem of DCA could be solved via a boosting technique and hence make DCA non-parametric. Thus, seeing the RL problem as a DC problem provides some interesting perspectives for future works.

**Acknowledgements**

The research leading to these results has received partial funding from the European Union Seventh Framework Program (FP7/2007-2013) under grant agreement number 270780 and the ANR ContInt program (MaRDi project, number ANR- 12-CORD-021 01). We also would like to thank professors Le Thi Hoai An and Pham Dinh Tao for helpful discussions about DC programming.

## Footnotes

[1]Others methods such as Approximate Linear Programming (ALP) [7, 8] or Dynamic Policy Programming (DPP) [4] address the same problem. Yet, they also rely on DP.

[2] This work could be easily extended to measurable state spaces as in [9]; we choose the finite case for the ease and clarity of exposition.

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
