[Supplementary Material · supplementary_file.pdf]

# Difference of Convex Functions Programming for Reinforcement Learning (Supplementary File)

**Bilal Piot**[1,2]**, Matthieu Geist**[1]**, Olivier Pietquin**[2,3]

[1]MaLIS research group (SUPELEC) - UMI 2958 (GeorgiaTech-CNRS), France
[2]LIFL (UMR 8022 CNRS/Lille 1) - SequeL team, Lille, France
[3] University Lille 1 - IUF (Institut Universitaire de France), France
bilal.piot@lifl.fr, matthieu.geist@supelec.fr, olivier.pietquin@univ-lille1.fr

This supplementary material provides the proofs of the results given in the main paper.

## 1 Proof of Th. 1

**Theorem 1.** *Let $\nu \in \Delta_{S \times A}$, $\mu \in \Delta_{S \times A}$, $\hat{\pi} \in A^S$ and $C_1(\nu, \mu, \hat{\pi}) \in [1, +\infty[\cup\{+\infty\}$ the smallest constant verifying $(1 - \gamma)\nu \sum_{t \geq 0} \gamma^t P_{\hat{\pi}}^t \leq C_1(\nu, \mu, \hat{\pi})\mu$, then:*

$$\forall Q \in \mathbb{R}^{S \times A}, \|Q^* - Q^\pi\|_{p,\nu} \leq \frac{2}{1-\gamma} \left( \frac{C_1(\nu, \mu, \pi) + C_1(\nu, \mu, \pi^*)}{2} \right)^{\frac{1}{p}} \|T^*Q - Q\|_{p,\mu}, \quad (1)$$

*where $\pi$ is greedy with respect to $Q$ and $\pi^*$ any optimal policy.*

*Proof.* This proof is widely inspired by the proof of a similar result for value functions (Th. 5.3 in [1]). First, we show that, componentwise:

$$\forall Q \in \mathbb{R}^{S \times A}, Q^* - Q^\pi \leq [(I - \gamma P_{\pi^*})^{-1} + (I - \gamma P_\pi)^{-1}]|T^*Q - Q|. \quad (2)$$

To do so, we use principally the fact that $T^*Q \geq T^{\pi^*}Q$ and $T^\pi Q = T^*Q$ as $\pi$ is greedy with respect to $Q$. Thus:

$$
\begin{aligned}
Q^* - Q^\pi &\underset{(a)}{=} T^{\pi^*}Q^* - T^*Q + T^*Q - T^\pi Q^\pi, \\
&\underset{(b)}{\leq} T^{\pi^*}Q^* - T^{\pi^*}Q + T^\pi Q - T^\pi Q^\pi, \\
&= \gamma P_{\pi^*}(Q^* - Q) + \gamma P_\pi(Q - Q^\pi), \\
&= \gamma P_{\pi^*}(Q^* - Q^\pi + Q^\pi - Q) + \gamma P_\pi(Q - Q^\pi),
\end{aligned}
$$

where equality $(a)$ comes from the fact that $Q^* = T^{\pi^*}Q^*$ and $Q^\pi = T^\pi Q^\pi$ and inequality $(b)$ from the fact that $T^*Q \geq T^{\pi^*}Q$ and $T^\pi Q = T^*Q$. Hence $(I - \gamma P_{\pi^*})(Q^* - Q^\pi) \leq \gamma(P_{\pi^*} - P_\pi)(Q^\pi - Q)$. In addition as $(I - \gamma P_{\pi^*})^{-1} = \sum_{t \geq 0} \gamma^t P_{\pi^*}^t$ is a matrix with only positive elements, we can multiply by $(I - \gamma P_{\pi^*})^{-1}$ on each side of the matricial inequality and conserve the order:

$$(Q^* - Q^\pi) \leq \gamma(I - \gamma P_{\pi^*})^{-1}(P_{\pi^*} - P_\pi)(Q^\pi - Q).$$

Moreover, we have:

$$
\begin{aligned}
(I - \gamma P_\pi)(Q^\pi - Q) &= \gamma P_\pi(Q - Q^\pi) + Q^\pi - Q, \\
&= T^\pi Q - T^\pi Q^\pi + Q^\pi - Q \underset{(c)}{=} T^*Q - Q,
\end{aligned}
$$

where equality $(c)$ comes from the fact that $Q^\pi = T^\pi Q^\pi$ and $T^\pi Q = T^* Q$. Therefore, we have:

$$
\begin{aligned}
(Q^* - Q^\pi) &\leq \gamma(I - \gamma P_{\pi^*})^{-1}(P_{\pi^*} - P_\pi)(Q^\pi - Q), \\
&= (I - \gamma P_{\pi^*})^{-1}(\gamma P_{\pi^*} - \gamma P_\pi)(I - \gamma P_\pi)^{-1}(T^* Q - Q), \\
&= (I - \gamma P_{\pi^*})^{-1}\left((I - \gamma P_\pi) - (I - \gamma P_{\pi^*})\right)(I - \gamma P_\pi)^{-1}(T^* Q - Q), \\
&= [(I - \gamma P_{\pi^*})^{-1} - (I - \gamma P_\pi)^{-1}](T^* Q - Q), \\
&\leq [(I - \gamma P_{\pi^*})^{-1} + (I - \gamma P_\pi)^{-1}]|T^* Q - Q|.
\end{aligned}
$$

Now, as it is mentioned in [1], in order to obtain an $L_{p,\mu}$ bound, we remark that if $u \in \mathbb{R}^{S \times A}$ and $v \in \mathbb{R}^{S \times A}$ are two vectors with positive elements and $N$ is stochastic matrix of size $N_S N_A \times N_S N_A$ such that $u \leq Nv$ and if $\nu \in \Delta_{S \times A}$ and $\mu \in \Delta_{S \times A}$ are two distributions such that $\nu N \leq C\mu$ where $C$ is a constant greater to $1$, then :

$$
\|u\|_{p,\nu} \leq C^{\frac{1}{p}}\|v\|_{p,\mu}. \tag{3}
$$

Indeed:

$$
\|u\|_{p,\nu}^p = \sum_{(s,a)\in S\times A}[u(s,a)]^p \nu(s,a) \underset{(d)}{\leq} \sum_{(s,a)\in S\times A}\left(\sum_{(s',a')\in S\times A} N((s,a),(s',a'))v(s',a')\right)^p \nu(s,a),
$$

$$
\underset{(e)}{\leq} \sum_{(s,a)\in S\times A}\sum_{(s',a')\in S\times A} N((s,a),(s',a'))[v(s',a')]^p \nu(s,a),
$$

$$
\underset{(f)}{\leq} C\sum_{(s',a')\in S\times A}\mu(s',a')[v(s',a')]^p = C\|v\|_{p,\mu}^p,
$$

where inequality $(d)$ is true because $u \leq Nv$, inequality $(e)$ is true using Jensen's Inequality and inequality $(f)$ comes from $\nu N \leq C\mu$.

To establish our bound (Eq. (1)), it is sufficient to remark that the inequality Eq. (2) can be written:

$$
Q^* - Q^\pi \leq A\frac{2}{1-\gamma}|T^* Q - Q|,
$$

where $A = \frac{1-\gamma}{2}[(I - \gamma P_{\pi^*})^{-1} + (I - \gamma P_\pi)^{-1}]$ is a stochastic matrix. Moreover by definition of $C_1(\nu,\mu,\pi)$ and $C_1(\nu,\mu,\pi^*)$ we have:

$$
\nu A \leq \left(\frac{C_1(\nu,\mu,\pi) + C_1(\nu,\mu,\pi^*)}{2}\right)\mu.
$$

Thus, if we rewrite Eq. (3), where $Q^* - Q^\pi$ plays the role of $u$, $\frac{2}{1-\gamma}|T^* Q - Q|$ plays the role of $v$, $A$ plays the role of $N$, and $\frac{C_1(\nu,\mu,\pi)+C_1(\nu,\mu,\pi^*)}{2}$ plays the role of $C$, then we have:

$$
\|Q^* - Q^\pi\|_{p,\nu} \leq \frac{2}{1-\gamma}\left(\frac{C_1(\nu,\mu,\pi) + C_1(\nu,\mu,\pi^*)}{2}\right)^{\frac{1}{p}}\|T^* Q - Q\|_{p,\mu}.
$$

$\square$

## 2 Proof of Th. 2

**Theorem 2.** *Let $\eta \in ]0,1[$ and $M$ be a finite deterministic MDP, with probability at least $1 - \eta$, we have:*

$$
\forall Q \in \mathfrak{Q}, \|T^* Q - Q\|_{p,\mu}^p \leq \|T^* Q - Q\|_{p,\mu_N}^p + \frac{2\|R\|_\infty}{1-\gamma}\sqrt{\varepsilon(N)},
$$

where $\varepsilon(N) = \frac{h(ln(\frac{2N}{h})+1)+ln(\frac{4}{\eta})}{N}$ and $h = 2N_A(d+1)$. *Moreover, with probability at least* $1 - 2\eta$:

$$\epsilon^{OBRM} = \|T^*Q_N - Q_N\|^p_{p,\mu} \leq \epsilon^B + \frac{2\|R\|_\infty}{1-\gamma}\left(\sqrt{\varepsilon(N)} + \sqrt{\frac{ln(1/\eta)}{2N}}\right),$$

*where* $\epsilon^B = \min_{Q \in \mathfrak{Q}} \|T^*Q - Q\|^p_{p,\mu}$ *is the error due to the choice of features.*

*Proof.* Here, we work with finite deterministic MDPs. This means that for each state-action couple $(s, a)$, there exists a unique next state $s'$. Let us note $l \in S^{S \times A}$ the function that maps each state-action couple $(s, a)$ to its next state $s'$. Then, we have:

$$\forall Q \in \mathbb{R}^{S \times A}, \forall (s, a) \in S \times A, T^*Q(s, a) = R(s, a) + \gamma \max_{b \in A} Q(l(s, a), b).$$

The result is based on Th 5.3 of [2], briefly recalled here. Let $\mathfrak{F} \subset \mathbb{R}^X$ be a set of measurable bounded real-valued functions where $X$ is a measurable set. In particular, we have $\forall f \in \mathfrak{F}, \forall x \in X, a \leq f(x) \leq b$ where $(a, b) \in \mathbb{R}^2$. Let $(x_i)_{i=1}^N$ be $N$ independent and identically distributed random variables taking their values in $X$ and such that $x_i \sim F$ where $F$ is a distribution over $X$. If $\mathfrak{F}$ has a finite VC-dimension (Vapnik-Chervonenkis dimension) $v(\mathfrak{F}) \leq h$ and $\eta \in ]0, 1[$ then with probability at least $1 - \eta$, we have:

$$\forall f \in \mathfrak{F}, \int_{x \in X} f(x)F(dx) \leq \frac{1}{N}\sum_{i=1}^N f(x_i) + (b-a)\sqrt{\varepsilon(N)},$$

where $\varepsilon(N) = \frac{h(ln(\frac{2N}{h})+1)+ln(\frac{4}{\eta})}{N}$. And with probability at least $1 - 2\eta$:

$$\min_{f \in \mathfrak{F}}\left(\int_{x \in X} f(x)F(dx)\right) \leq \min_{f \in \mathfrak{F}}\left(\sum_{i=1}^N f(x_i)\right) + (b-a)\left(\sqrt{\varepsilon(N)} + \sqrt{\frac{ln(1/\eta)}{2N}}\right).$$

Our result has exactly the same form where $X = S \times A$, the random variables $(x_i)_{i=1}^N$ are replaced by $(S_i, A_i)_{i=1}^N$, the distribution $F = \mu \in \Delta_{S \times A}$, the space $\mathfrak{F}$ is replaced by $\tilde{\mathfrak{Q}} = \{|T^*Q - Q|^p, \text{ where } Q \in \mathfrak{Q}\}$, $a = 0$ and $b = \frac{2\|R\|_\infty}{1-\gamma}$. The only thing left to prove our result is to show that the VC-dimension of $\tilde{\mathfrak{Q}}$, $v(\tilde{\mathfrak{Q}})$, is such that $v(\tilde{\mathfrak{Q}}) \leq 2N_A(d+1)$.

First, let recall some definitions relative to the VC-dimension of a set of functions. Let $f \in \mathbb{R}^X$ be a real-valued function where $X$ is a set and $(x_i, t_i)_{i=1}^N \in (X \times \mathbb{R})^N$ a sequence of couples of one element of $X$ and one real value, $m(f, x_i, t_i) = \mathbf{1}_{\{f(x_i) \geq t_i\}}$ is a boolean which says if $f(x_i)$ is greater than $t_i$ or not. Moreover, $M(f, (x_i, t_i)_{i=1}^N) = (m(f, x_i, t_i))_{i=1}^N$ can be seen as a boolean vector of size $N$ and we call it the message relative to both the function $f$ and the sequence $(x_i, t_i)_{i=1}^N$. Let $\mathfrak{F} \subset \mathbb{R}^X$, $\mathfrak{N}(\mathfrak{F}, (x_i, t_i)_{i=1}^N)$ is the number of possible messages $M(f, (x_i, t_i)_{i=1}^N)$ obtained when $f \in \mathfrak{F}$:

$$\mathfrak{N}(\mathfrak{F}, (x_i, t_i)_{i=1}^N) = \text{Card}(\{M(f, (x_i, t_i)_{i=1}^N), f \in \mathfrak{F}\}),$$

where Card denotes the cardinal of a given set. As $M(f, (x_i, t_i)_{i=1}^N)$ is a boolean vector of size $N$, we have $\mathfrak{N}(\mathfrak{F}, (x_i, t_i)_{i=1}^N) \leq 2^N$. In addition, we define $\mathfrak{N}(\mathfrak{F}, N) = \sup_{(x_i, t_i)_{i=1}^N \in (X \times \mathbb{R})^N} \mathfrak{N}(\mathfrak{F}, (x_i, t_i)_{i=1}^N)$ the maximum number of possible messages when $f \in \mathfrak{F}$ that a given sequence $(x_i, t_i)_{i=1}^N$ can produce. Finally, the VC-dimension of $\mathfrak{F}$ is defined by:

$$v(\mathfrak{F}) = \inf_{N \in \mathbb{N}}\{\mathfrak{N}(\mathfrak{F}, N) < 2^N\}.$$

In our proof, the followings properties relative to VC-dimensions of functions sets are needed.

**Property 1.** *Let* $(\mathfrak{F}_k)_{k=1}^K$ *be a sequence of set of functions where* $\mathfrak{F}_k \subset \mathbb{R}^X$ *and* $v(\mathfrak{F}_k)$ *is finite. Then, the set of functions* $\mathfrak{F} = \{\max_{k \in [|1:K|]} f_k, \text{ where } \forall k \in [|1:K|], f_k \in \mathfrak{F}_k\}$ *has a finite VC-dimension lower that* $\sum_{k=1}^K v(\mathfrak{F}_k)$.

Let $(x_i, t_i)_{i=1}^N \in (X \times \mathbb{R})^N$, $(f_k \in \mathfrak{F}_k)_{k=1}^K$ and $f = \max_{k \in [|1:K|]} f_k$, then:

$$M(f, (x_i, t_i)_{i=1}^N) = M(f_1, (x_i, t_i)_{i=1}^N) \vee M(f_2, (x_i, t_i)_{i=1}^N) \cdots \vee M(f_K, (x_i, t_i)_{i=1}^N),$$

where $\vee$ is the boolean disjunction (the inclusive or). Thus, the number of possible messages $\mathfrak{N}(\mathfrak{F}, (x_i, t_i)_{i=1}^N)$ is such that:

$$\mathfrak{N}(\mathfrak{F}, (x_i, t_i)_{i=1}^N) \leq \prod_{i=1}^K \mathfrak{N}(\mathfrak{F}_k, (x_i, t_i)_{i=1}^N).$$

This implies that:

$$\mathfrak{N}(\mathfrak{F}, N) \leq \prod_{i=1}^K \mathfrak{N}(\mathfrak{F}_k, N).$$

Now, if we choose $N$ such that $N > \max_{k \in [|1:K|]} v(\mathfrak{F}_k)$, we have:

$$\mathfrak{N}(\mathfrak{F}, N) \leq \prod_{i=1}^K 2^{v(\mathfrak{F}_k)} = 2^{\sum_{k=1}^K v(\mathfrak{F}_k)}.$$

So, $\mathfrak{F}$ has a finite VC-dimension lower that $\sum_{k=1}^K v(\mathfrak{F}_k)$. A second interesting result is a corollary of the previous proposition.

**Property 2.** *Let $\mathfrak{F} \subset \mathbb{R}^X$ be a set of functions with a finite VC-dimension, then the set of functions $\mathfrak{F}_{|.|} = \{|f|, f \in \mathfrak{F}\}$ has a VC-dimension lower than $2v(\mathfrak{F})$.*

Indeed, to prove the result, we remark that $|f| = \max(f, -f)$ and as the set $\mathfrak{F}_- = \{-f, f \in \mathfrak{F}\}$ has the same VC-dimension than $\mathfrak{F}$, we apply the previous result to conclude. The last result needed is the following.

**Property 3.** *Let $\mathfrak{F} \subset \mathbb{R}_+^X$ be a set of functions with a finite VC-dimension, then the set $\mathfrak{F}_p = \{f^p, f \in \mathfrak{F}\}$, where $p \geq 1$, has the same VC-dimension.*

To show this property, let $f \in \mathfrak{F}$ and $(x_i, t_i)_{i=1}^N \in (X \times \mathbb{R})^N$, then:

$$M(f, (x_i, t_i)_{i=1}^N) = M(f^p, (x_i, \text{sgn}(t_i)|t_i|^p)_{i=1}^N),$$

where sgn is the sign function, thus $\mathfrak{N}(\mathfrak{F}, (x_i, t_i)_{i=1}^N) = \mathfrak{N}(\mathfrak{F}_p, (x_i, \text{sgn}(t_i)|t_i|^p)_{i=1}^N)$. As the function $t \to \text{sgn}(t)|t|^p$ is a bijection, then:

$$\sup_{(x_i, t_i)_{i=1}^N \in (X \times \mathbb{R})^N} \mathfrak{N}(\mathfrak{F}_p, (x_i, t_i)_{i=1}^N) = \sup_{(x_i, t_i)_{i=1}^N \in (X \times \mathbb{R})^N} \mathfrak{N}(\mathfrak{F}_p, (x_i, \text{sgn}(t_i)|t_i|^p)_{i=1}^N).$$

So, $\mathfrak{N}(\mathfrak{F}, N) = \mathfrak{N}(\mathfrak{F}_p, N)$ which implies that $v(\mathfrak{F}) = v(\mathfrak{F}_p)$.

Now, let show that the VC-dimension of $\tilde{\mathfrak{Q}}$ is such that $v(\tilde{\mathfrak{Q}}) \leq 2N_A(d+1)$. To do so, we are going to proceed in several steps. The first step is to remark that:

$$T^* Q_\theta(s, a) - Q_\theta(s, a) = \max_{b \in A} \left( \sum_{k=1}^d \theta_k [\gamma \phi_k(l(s, a), b) - \phi_k(s, a)] + R(s, a) \right).$$

Thus, if we note $\forall b \in A, \forall k \in [|1 : d|], \psi_k^b(s, a) = \gamma \phi_k(l(s, a), b) - \phi_k(s, a)$, we have:

$$T^* Q_\theta(s, a) - Q_\theta(s, a) = \max_{b \in A} [\sum_{k=1}^d \theta_k \psi_k^b(s, a) + \theta_0 R(s, a)].$$

where $\theta_0 = 1$. Let $b \in A$, the set of functions $\mathfrak{F}_b = \{f_\theta = \sum_{k=1}^d \theta_k \psi_k^b + \theta_0 R, \theta \in \mathbb{R}^d\}$ has a finite VC-dimension lower than $d + 1$ as the functions $f_\theta \in \mathfrak{F}_b$ depends linearly on $d + 1$ parameters [2] where one of them ($\theta_0$) is fixed. Now, we want to show that the set $\mathfrak{F} = \{f_\theta = \max_{b \in A} [\sum_{k=1}^d \theta_k \psi_k^b + R], \theta \in \mathbb{R}^d\}$ has a finite VC-dimension lower than $N_A(d+1)$. To do so, we remark that $\mathfrak{F} = \{f = \max_{b \in A} f_b, \text{ where } \forall b \in A, f_b \in \mathfrak{F}_b\}$, thus by applying property. 1 to $\mathfrak{F}$ we obtain that its VC-dimension is lower than $N_A(d+1)$. Now, let define the

set of functions $\mathfrak{F}_{|.|} = \{f_\theta = |\max_{b\in A}[\sum_{k=1}^d \theta_k \psi_k^b + R]|, \theta \in \mathbb{R}^d\} = \{|T^*Q_\theta - Q_\theta|, \theta \in \mathbb{R}^d\}$.
We remark that, $\mathfrak{F}_{|.|} = \{|f|, f \in \mathfrak{F}\}$, thus, by using property 2, the VC-dimension of $\mathfrak{F}_{|.|}$ is lower than $2N_A(d+1)$. Finally, we define the set $\mathfrak{F}_p = \{|T^*Q_\theta - Q_\theta|^p, \theta \in \mathbb{R}^d\}$, where $p \geq 1$. We have $\mathfrak{F}_p = \{f, f \in \mathfrak{F}_{|.|}\}$, thus, by applying property 3, we have that the VC-dimension of $\mathfrak{F}_p$ is lower than $2N_A(d+1)$. As $\tilde{\mathfrak{Q}} \subset \mathfrak{F}_p$, we have $v(\tilde{\mathfrak{Q}}) \leq 2N_A(d+1)$. $\qquad\square$

## 3    Proof of Th.4

**Theorem 3.** *There exists explicit polyhedral decompositions of $J_{p,\mu_N}^p$ when $p = 1$ and $p = 2$. For $p = 1$:*
$$J_{1,\mu_N} = G_{1,\mu_N} - H_{1,\mu_N},$$
*where $G_{1,\mu_N} = \frac{1}{N}\sum_{i=1}^N 2\max(g_i, h_i)$ and $H_{1,\mu_N} = \frac{1}{N}\sum_{i=1}^N (g_i + h_i)$, with $g_i = \langle\phi(S_i, A_i), .\rangle + R(S_i, A_i)$ and $h_i = \gamma \sum_{s'\in S} P(s'|S_i, A_i)\max_{a\in A}\langle\phi(s', a), .\rangle$. For $p = 2$:*
$$J_{2,\mu_N}^2 = G_{2,\mu_N} - H_{2,\mu_N},$$
*where $G_{2,\mu_N} = \frac{1}{N}\sum_{i=1}^N [\bar{g}_i^2 + \bar{h}_i^2]$ and $H_{2,\mu_N} = \frac{1}{N}\sum_{i=1}^N [\bar{g}_i + \bar{h}_i]^2$ with:*

$$\bar{g}_i = \max(g_i, h_i) + g_i - \left(\langle\phi(S_i, A_i) + \gamma \sum_{s'\in S} P(s'|S_i, A_i)\phi(s', a_1), .\rangle - R(S_i, A_i)\right),$$

$$\bar{h}_i = \max(g_i, h_i) + h_i - \left(\langle\phi(S_i, A_i) + \gamma \sum_{s'\in S} P(s'|S_i, A_i)\phi(s', a_1), .\rangle - R(S_i, A_i)\right).$$

*Proof.* When $p = 1$, it is sufficient to remark that for two functions $g, h \in \mathbb{R}^E$, $|g - h| = 2\max(g, h) - (g + h)$. Thus, let $G_{1,\mu_N} = \frac{1}{N}\sum_{i=1}^N 2\max(g_i, h_i)$ and $H_{1,\mu_N} = \frac{1}{N}\sum_{i=1}^N (g_i + h_i)$ which are convex and continuous (as a finite maximum of convex and continuous functions and a positively weighted sum of convex and continuous functions are convex and continuous), then $J_{1,\mu_N} = G_{1,\mu_N} - H_{1,\mu_N}$. When $p = 2$, the decomposition is less straightforward. An important property that we use is the fact that $f^2$ is a convex and continuous functions if $f$ is a positive and continuous convex function. The first thing to do is to find a decomposition of $f_i = \bar{g}_i - \bar{h}_i$ such that $\bar{g}_i$ and $\bar{h}_i$ are positive and continuous convex functions. To do so, it is sufficient to remark that:

$$\bar{g}_i = \max(g_i, h_i) + g_i - \left(\langle\phi(S_i, A_i) + \gamma \sum_{s'\in S} P(s'|S_i, A_i)\phi(s', a_1), .\rangle - R(S_i, A_i)\right),$$

$$\bar{h}_i = \max(g_i, h_i) + h_i - \left(\langle\phi(S_i, A_i) + \gamma \sum_{s'\in S} P(s'|S_i, A_i)\phi(s', a_1), .\rangle - R(S_i, A_i)\right).$$

are positive and continuous convex functions. Thus:

$$J_{2,\mu_N}^2 = \frac{1}{N}\sum_{i=1}^N [\bar{g}_i - \bar{h}_i]^2 = \frac{1}{N}\sum_{i=1}^N [\bar{g}_i^2 + \bar{h}_i^2] - \frac{1}{N}\sum_{i=1}^N [\bar{g}_i + \bar{h}_i]^2.$$

As $\bar{g}_i$ and $\bar{h}_i$ are convex, continuous and positive then $\bar{g}_i^2 + \bar{h}_i^2$ and $[\bar{g}_i + \bar{h}_i]^2$ are convex and continuous. So, if we note $G_{2,\mu_N} = \frac{1}{N}\sum_{i=1}^N [\bar{g}_i^2 + \bar{h}_i^2]$ and $H_{2,\mu_N} = \frac{1}{N}\sum_{i=1}^N [\bar{g}_i + \bar{h}_i]^2$ which are convex and continuous, we have $J_{2,\mu_N}^2 = G_{2,\mu_N} - H_{2,\mu_N}$. We also remark that $G_{2,\mu_N}$, $H_{2,\mu_N}$, $G_{1,\mu_N}$ and $H_{2,\mu_N}$ are polyhedral and $J_{p,\mu_N}^p$ is under bounded by 0, thus DCA has better convergence properties than in the classical case. $\qquad\square$