[Reviews · NeurIPS 2014]

Submitted by Assigned_Reviewer_4

The paper presents a new technique for solving MDPs. The new technique, presented as an alternative to approximate policy/value iteration, consists in directly minimizing the Optimal Bellman Residual (OBR). The authors first motivate their method by showing that the loss bound of OBR is often tighter than the loss bound of policy/value iteration, which is a known result [9,15]. The authors then show that an empirical estimate of OBR is consistent in the Vapnick sense, i.e. minimizing the empirical OBR is equivalent to minimizing an upper bound on the true OBR, which is unknown when the MDP model is unknown. Finally, the authors show that OBR can be decomposed into a difference of two convex functions, and a standard Difference of Convex Functions (DC) optimization method can be used for finding a local optimum. The authors argue that the obtained optimum is usually not very different from the global one. Finally, the authors show that the performance of their method is similar to the performance of standard techniques, such as LSPI and Fitted-Q learning. The authors also show that the values obtained from OBR minimization have a smaller variance.

Pros: The paper is very clear, well-written and well-organized. Technically, the paper is strong and sound. The proofs are not trivial and quite original. Theorem 4 (decomposition of OBR into a difference of convex functions) is probably the most useful result in this paper from a practical point of view. The motivation from a loss bound (Sec 2.2) is appreciated. Moreover, the authors prove the consistency of empirical risk minimization when the value function is approximated with basis functions (features).

Cons: The empirical evaluation is very weak. The authors experiment only on a simple artificial MDP, and their method is not better than LSPI or Fitted-Q learning. If it will turn out that OBR minimization is never better than LSPI, then this whole contribution becomes questionable. Moreover, the authors did not report the computational effort needed for solving multiple convex optimization problems. My intuition is that if the DC decomposition is not well-done, than DC programming can be slower than DC programming.
However, I agree that the main contribution of this paper is pointing to this new way of solving MDPs, which seems intriguing. I would like to see how the DC iterations are related to the policy/value iterations.

Questions/comments:

1) What is \mu in line 120? How is it different from the state-action distribution v? This should be introduced before Equation (1).
2) Equation (3) is mostly a repetition of Equation (2)
3) Typos in line 158: "better that", "manly", in line 313 "is currently is"
4) The proof of Theorem 3 is straightforward, you can cut it off, remove Equation (3) and bring the more important proof of Theorem 4 to the main paper.
5) In your experiment, is the computational cost of DC programming in the same order of magnitude as dynamic programing?

Summary: Overall, this is a very strong, well-written, paper. Up to my knowledge, the decomposition of the optimal Bellman residual into a difference of convex functions is original. The empirical evaluation is weak and non-conclusive, but I don't think that this point should be held against the paper.

Submitted by Assigned_Reviewer_5

I think the paper is well-written and that the authors do a fine job of introducing us to the DC programming area and its potential relevance to reinforcement learning. Its hard to know how helpful this will eventually be, however. DC programming itself is not known to be computationally efficient. Further, the consistency results in Section 3 seem to be quite limited in scope (even the one the authors discuss that applies to a class of stochastic MDPs). The computational results show some potential, though for simple and very low-dimensional (2d) problems.

Some minor notes:

o In the last paragraph of page 3: "manly" --> "mainly" and why the "+" in one "+\inf" but not the other?

o Second paragraph of page 4: should "too important" be "too large"?

o Section 6 mentions the possibility of boosting as though it is something that may differentiate DCA. Why can't boosting also be used with other RL algorithms?
Summary: Early-stage work introducing a new algorithmic approach to the area of reinforcement learning. Not clear yet how useful this will be eventually.

Submitted by Assigned_Reviewer_43

Summary:
The paper is focused on the application of Difference of Convex Functions algorithms to minimize the norm of the Optimal Bellman Residual. In order to support such an application, the author(s) establish that the norm of the OBR can be represented as a difference of convex functions, and that this loss function has desirable theoretical benefits to the error bounds.

The decomposition that proves the OBR can be written as a difference of convex functions is straightforward enough to follow without further proof and establishes an interesting connection. I have not seen this shown before, and so despite seeming obvious in hindsight provides a novel contribution.

A small empirical study comparing the use of DCA for minimizing the norm of the OBR with other reinforcement learning algorithms (LSPI and Fitted-Q). These results are over a collection of randomly generated finite-state MDPs and show comparable results, but with lower variance.

Discussion:

The paper presents an interesting application of algorithms for non-convex optimization to the reinforcement learning problem. In particular, the authors do a good job of motivating the use of the norm of the OBR and for its decomposition into a difference of convex functions. This is an interesting new direction to consider and these initial theoretical and empirical results suggest it is worth further exploration.

Although I was left with many questions about this approach, especially about the details of the empirical study, the paper as a whole is a good contribution to the field.

Recommendations:

The empirical study is very limited. At the least I would like to see another application domain and a thorough discussion of the methodology behind the study. Specifically, the algorithms for DCA, LSPI, and Fitted-Q all have tunable parameters and the author(s) do not provide any information about the values or how they were chosen. There are a number of typos in the math and writing. Although these do not greatly detract from the paper, the authors should be sure to fix them.

Summary: Gives a novel approach to value-function based reinforcement learning. Could be improved, especially in the empirical study, but as it is this work provides an interesting contribution.
Author Feedback
Author rebuttal: We would like to thank the reviewers for their helpful comments which will definitely help us to improve our paper. Most comments of each reviewer are positive and it seems that the idea proposed has been found interesting. We are aware that the experimental results are quite limited, although representative of many problems, yet we wanted to stress on the main contributions of the paper which is DCA for RL. We will definitely follow the reviewer's comments to improve the paper and will correct the listed typos.